# Could Vitamins Help in the Fight Against COVID-19?

**DOI:** 10.3390/nu12092550

**Published:** 2020-08-23

**Authors:** Thomas H Jovic, Stephen R Ali, Nader Ibrahim, Zita M Jessop, Sam P Tarassoli, Thomas D Dobbs, Patrick Holford, Catherine A Thornton, Iain S Whitaker

**Affiliations:** 1Reconstructive Surgery & Regenerative Medicine Research Group, Institute of Life Sciences, Swansea University Medical School, Swansea University, Swansea SA28PY, UK; Nader.Ibrahim@wales.nhs.uk (N.I.); Zita.Jessop@wales.nhs.uk (Z.M.J.); sam.tarassoli@wales.nhs.uk (S.P.T.); Thomas.Dobbs@wales.nhs.uk (T.D.D.); 2Welsh Centre for Burns & Plastic Surgery, Morriston Hospital, Swansea SA66NL, UK; 3Institute for Optimum Nutrition, Ambassador House, Paradise Road, Richmond TW9 1SQ, UK; pat@patrickholford.com; 4Institute of Life Sciences 1, Swansea University Medical School, Swansea University, Swansea SA2 8PY, UK; c.a.thornton@swansea.ac.uk

**Keywords:** COVID-19, SARS-CoV-2, vitamin, immunonutrition, supplementation

## Abstract

There are limited proven therapeutic options for the prevention and treatment of COVID-19. The role of vitamin and mineral supplementation or “immunonutrition” has previously been explored in a number of clinical trials in intensive care settings, and there are several hypotheses to support their routine use. The aim of this narrative review was to investigate whether vitamin supplementation is beneficial in COVID-19. A systematic search strategy with a narrative literature summary was designed, using the Medline, EMBASE, Cochrane Trials Register, WHO International Clinical Trial Registry, and Nexis media databases. The immune-mediating, antioxidant and antimicrobial roles of vitamins A to E were explored and their potential role in the fight against COVID-19 was evaluated. The major topics extracted for narrative synthesis were physiological and immunological roles of each vitamin, their role in respiratory infections, acute respiratory distress syndrome (ARDS), and COVID-19. Vitamins A to E highlighted potentially beneficial roles in the fight against COVID-19 via antioxidant effects, immunomodulation, enhancing natural barriers, and local paracrine signaling. Level 1 and 2 evidence supports the use of thiamine, vitamin C, and vitamin D in COVID-like respiratory diseases, ARDS, and sepsis. Although there are currently no published clinical trials due to the novelty of SARS-CoV-2 infection, there is pathophysiologic rationale for exploring the use of vitamins in this global pandemic, supported by early anecdotal reports from international groups. The final outcomes of ongoing trials of vitamin supplementation are awaited with interest.

## 1. Introduction

December 2019 saw clusters of patients in the city of Wuhan, Hubei Provence, China presenting with a severe acute respiratory syndrome coronavirus (SARS-CoV)-like illness. The causative organism, now termed SARS-CoV-2 and its correspondent disease, COVID-19, has quickly spread from Asia via Europe to the rest of the world. Previous viral pandemics due to novel corona and influenza viruses, such as SARS-CoV, Middle Eastern respiratory syndrome coronavirus (MERS-CoV), and H1N1 influenza A have been a warning to global healthcare. However, none of these pathogens have had such a catastrophic impact worldwide as the novel coronavirus SARS-CoV-2, a positive-sense single-stranded enveloped RNA virus, which is the seventh coronavirus known to infect humans [1]. RNA viruses are known to possess very high mutation rates, which correlate with enhanced virulence and evolvability [2]. COVID-19 highlights the potency of this, with a case-fatality rate of 2.3% and over 5,000,000 infections in 216 countries at the time of writing [3,4]. The pandemic is gathering speed across the world despite increasingly more drastic non-pharmacological interventions to limit its spread. The clinical spectrum of COVID-19 ranges from asymptomatic carriage, mild upper respiratory tract infection (URTI), severe viral pneumonia to acute respiratory distress syndrome (ARDS) and death [5]. Research has led to identification of the angiotensin-converting enzyme (ACE) 2 as the cell-entry receptor for SARS-CoV-2 [6]. Despite this discovery, a systematic study of virus dynamics and the host response is yet to be completed [7].

Although vaccine development is proceeding at an unprecedented pace, with reports of phase 1 trials already in progress [8], given the length of time required to develop, evaluate, produce, and disseminate it to 7.8 billion citizens worldwide, there is an urgent need for evidence-based treatment modalities for SARS-CoV-2.

At present, no effective antiviral therapy has been confirmed and symptomatic supportive intervention is still the main treatment. There has been a previous suggestion that there is a role for vitamin supplementation to attenuate the severity of the common cold, the development of ARDS, and augment the immune system via antioxidant properties. The role of vitamin and mineral supplementation or “immunonutrition” has previously been explored in a number of trials in intensive care settings.

This review aims to interrogate the current evidence base, and to present the potential immune-mediating, antioxidant, and antimicrobial roles of vitamins A to E in the context of respiratory disease, and to extrapolate this evidence to evaluate the potential roles in the fight against COVID-19.

## 2. Methods

A systematic search strategy was employed to identify clinical, animal, and in vitro studies that may elucidate mechanisms by which vitamins may play a role in the fight against COVID-19. The Medline (1946-present), EMBASE (1980-present), Cochrane Trials Register, Clinicaltrials.gov, and WHO International Clinical Trial Registry were all searched using the developed search strategy up to 11 May 2020, supplemented with a search of newspaper and internet article sources using the media database, Nexis.

In total, 5506 manuscripts were screened for relevance, of which 367 underwent full-text review, with 204 studies warranting discussion in this narrative review (Figure 1). Figure 2 summarizes the Nexis search data and highlights the increasing media interest in the role of vitamin supplementation in the coronavirus pandemic since December 2019.

### Inclusion and Exclusion Criteria

Case reports, case cohorts, case–control, randomized controlled studies, and meta-analyses were considered for inclusion. Specifically, studies involving humans or animals (including ex vivo and in vitro models) with a clinical (any recognized diagnostic criteria) diagnosis of SARS-CoV, SARS-CoV-2 infection, MERS-CoV, or viral-induced ARDS receiving vitamin supplementation of any form were included. In addition to non-English language articles, lipopolysaccharide (LPS) animal models of ARDS and clinical studies where the etiology of ARDS included sepsis were avoided. This was because the clinical course of the viral pneumonia seen in COVID-19 has been described as an entity distinct from the Third International Consensus Definitions for Sepsis and Septic Shock. There was a paucity of published original evidence pertaining to a role of vitamin supplementation in the current coronavirus pandemic, or previous MERS and SARS pandemics. As such, studies on the role of vitamins in respiratory and immune system physiology and non-coronavirus respiratory tract infections were identified and scrutinized for their potential relevance to the current coronavirus pandemic.

## 3. Immunonutrition

Immunonutrition refers to the modulation of the immune system through the modification of dietary nutrients [9]. Due to the proinflammatory state in ARDS it has been postulated for many years that increasing the level of antioxidant nutrients within the body will have a beneficial effect [10]. Furthermore, improving lymphocyte, macrophage, and neutrophil function through the addition of nutrients, such as glutamine, has also been described as being beneficial [11]. Dietary supplementation of nutrients has been recognized to alter the clinical course in a wide range of patients, including those who are critically ill [12]. There are a number of commercially available enteral and parenteral formulas containing a mix of immunonutrients. These most commonly consist of antioxidant vitamins, trace elements, essential amino acids, or fatty acids and gamma-linolenic acid [13].

The evidence supporting the beneficial effects of immunonutrition in acute lung injury (ALI) and ARDS is contradictory and often limited by significant bias. Clinical studies in patients with ARDS have shown that their baseline plasma levels of beta-carotene, retinol, alpha-tocopherol, and total radical antioxidant potential were all lower than normal. This could be normalized after 4 days of feeding with eicosapentaenoic acid, gamma-linolenic acid, and antioxidants, compared to a control group not receiving supplementation. There were, however, no clinical outcomes reported [14]. In a further randomized study investigating a similar supplementation of eicosapentaenoic acid, gamma-linolenic acid, and antioxidants (including vitamins) in patients with severe sepsis and septic shock, a significant reduction in 28-day mortality, ventilator-free days, and intensive care unit-free days was observed [15]. Despite these results, the systematic review and meta-analysis data of those reported studies is less promising. In a systematic review of enteral immunomodulatory diets in those with ALI and ARDS, it was found that in a pooled analysis of the six controlled trials included, there was no significant reduction in all-cause mortality and no increase in the number of ventilator-free or intensive care-free days [16]. There was a suggestion from the two studies that those at highest risk of mortality might, however, benefit. A 2019 Cochrane review identified 10 studies including 1015 ARDS patients randomized to dietary supplementation of omega-3 fatty acids and antioxidants. There was no reduction in all-cause mortality. There is a suggestion that ventilator-free days and intensive care unit-free days were reduced, although the authors conclude that this is uncertain due to the low quality of the evidence included [17].

The importance of immunonutrition during the current SARS-CoV-2 pandemic has been identified by others, with the publication of protocols for the early nutritional supplementation of those with the disease [18]. Reported work on this during this pandemic is limited. In many trials of dietary supplementation, the ‘immunonutrition diet’ is heterogenous, consisting of several vitamins, minerals, and fatty acids administered together, making it difficult to ascertain the potential value of each vitamin as discussed below.

### 3.1. Vitamin A

#### 3.1.1. Source and Physiological Role

Vitamin A is, by structural definition, all-trans-retinol, a retinol in which all four exocyclic double bonds have an E- (also known as trans) geometry (Figure 3) [19,20,21]. Retinoids, a term coined in the mid-1970s, compromises both natural and synthetic chemical species that have similar structural appearances with or without biological component/activity, the biological species being those we colloquially refer to as vitamin A [22]. This would make vitamin A, a natural retinoid species. The acquisition of retinoids is generally from a dietary source in a preformed form or provitamin A carotenoids. The most active retinoid has been found to be retinoic acid. Retinoic acid has been discovered to regulate the transcription of more than 500 genes by its binding mechanism: RARα/β/γ receptor to its retinoid X counterpart [23].

The sources of these are found commonly in animal food sources, such as meat, fish, and eggs. The carotenoids, generally in the form of alpha/beta/gamma carotene, are more likely to found in fruit and vegetables; β-carotene specifically contributes to the orange color of food and is typically associated with carrots and sweet potatoes [24]. These provitamin carotenoids are converted to retinoids in the body [25].

The site of action is throughout the tissues, where both retinol and β-carotene are oxidized to retinal and retinoic acid, which are essential for the variety of biological roles of vitamin A [26]. The liver plays a key role: Retinol is esterified to retinyl esters and stored in the stellate cells [27]. The oxidative ability of vitamin A has been the subject of much debate, with cases made for it being both antioxidative as well as pro-oxidative [28].

The biological functions of retinoids are wide and varied, including gene transcription [29], vision maintenance and health (in the form of retinal) [30], epithelial and membrane regulation (from skin to mucous to teeth) to bone metabolism, and antioxidative properties [31,32]. However, it has a major role in immune system modulation [19]. Although not fully understood, many studies provide evidence that vitamin A plays a crucial role in certain immunoregulatory processes. Vitamin A has been found to not only promote proliferation of T-lymphocytes (through the increase of IL-2) but to also promote their differentiation, especially into regulatory T cells. This has been interestingly used as an adjunct to vaccine use, including tetanus, diphtheria, measles, influenza, rabies, and malaria, in infants to improve the antibody response [33,34,35,36].

#### 3.1.2. Mechanism of Action in Disease

Early work by Paiva et al. determined that vitamin A supplementation improved pulmonary function test results in patients with chronic obstructive pulmonary disease (COPD) [37]. This was explained on a cellular level by studies that showed that in fetal lung explants, retinoic acid (the most active retinoid species) was able to control the expression of surfactant protein [38,39,40]. Pediatric asthma studies (alongside supportive in vitro work) of vitamin A deficiency reinforced these theories by demonstrating that increased serum vitamin A induced good pulmonary function as retinoic acid can reverse airway hyper-responsiveness, in turn conferring protection from asthma by downregulation of oxidative stress [38,40].

#### 3.1.3. Respiratory Infections

Since 1928, vitamin A has been found to have antimicrobial qualities, tied to the immune-modulating factor of its derivatives [39,41]. Later studies showed that the risk of respiratory disease was ‘more closely associated to vitamin A status then with general nutritional status [42]. The deficiency of retinol has also been discovered to cause squamous metaplasia of the first barrier of respiratory defense, the epithelium [43]. On a clinical basis, addressing vitamin A deficiency has effectively prevented tuberculosis (TB) in high-risk patients [44]. Though it should be noted that wide-scale studies have not shown a beneficial effect in areas of high prevalence of vitamin A deficiency [45]. Vitamin A is believed to have antioxidative properties, and has a key role in the development of bronchopulmonary dysplasia and neonatal respiratory distress syndrome [46,47]. This finding was pioneered by Hustead et al. in 1984 with his work on retinol concentrations in preterm infants, in which it was discovered that neonates developing bronchopulmonary dysplasia had lower concentrations of retinol at birth [48].

#### 3.1.4. Relevance to COVID-19

The pulmonary, immunomodulatory, and antimicrobial roles of vitamin A may enact a crucial element in the fight against viral diseases, including COVID-19, and are summarized in Figure 4 [18]. From a pulmonary perspective, retinoic acid has been implicated in modulating the pathogenesis of ARDS, influencing the production of IL1-β and IL-1 receptor antagonist by alveolar macrophages, and the subsequent pulmonary infiltration of neutrophils [49]. In addition, a study of retinoic acid with simvastatin has demonstrated augmented pulmonary regeneration and remodeling in animal studies [50], meaning both oxidative damage and the regenerative capacity of the lungs may be, in part, mediated through vitamin A-dependent mechanisms. There may also be relevance in the role of vitamin A in viral infections. Retinoids have been implicated in the development of an innate immunity against measles virus in vitro through an interferon-mediated mechanism, which in particular renders bystander cells protected against a subsequent round of viral replication [51]. The vitamin has been the subject of animal studies looking at inactivated bovine coronaviruses, with dietary supplements shown to increase the effect of antibody responses to the vaccine in feedlot calves [52], and lower levels of vitamin A in chickens with viral infections has been associated with an increased rate of epithelial damage to tissues [53]. These findings correlate with clinical studies, which have found increasing host susceptibility to influenza and SARS-CoV with a lower concentration of vitamin A in several disease models [54]. In light of its pulmonary and immunological roles, oral supplementation of vitamin A is currently being investigated in the treatment of COVID-19 alongside a host of other antioxidants [55].

### 3.2. Vitamins B

#### 3.2.1. Sources and Physiological Role

B vitamins are a class of water-soluble vitamins (B_1_, B_2_, B_3_, B_5_, B_6_, B_7_, B_9_, and B_12_) that play important roles in cell metabolism (Table 1). They are chemically distinct entities but may coexist in the same foods, including meat and plant-based sources. B12 is found predominantly in meat, such as turkey, tuna, and liver, whereas folate is largely present in plant products, such as legumes (pulses or beans), greens, nuts, whole grains, potatoes, bananas, chili peppers, tempeh, and yeast. They are also found in small quantities in unprocessed carbohydrates whereas processed carbohydrates, such as sugar and white flour, have lower vitamin B levels, often corrected by supplementation. Dietary supplements containing all eight are referred to as a vitamin B complex.

Studies suggest that vitamin B complex may regulate cytokine/chemokine generation and mediate interaction with immune cells involved in pathophysiological pathways and inflammation [56]. Vitamin B_1_ (thiamine), like other B-complex vitamins, acts as a coenzyme in its phosphorylated forms, contributing to glucose, fat, and protein metabolism, participating in the production of energy for the body. Its deficiency in the nervous system may impair the ability to synthesize fatty acid and cholesterol, necessary for membrane function. With specific regard to inflammation, vitamin B_1_ deficiency in the brain induces overexpression of proinflammatory mediators, such as IL-1, IL-6, COX-2, and TNF-α, which cause neuronal cell death in the central nervous system (CNS) damage and neuroinflammation, leading to Wernicke’s encephalopathy and an irreversible dementia of Korsakoff’s syndrome [57]. Vitamin B_2_ (riboflavin) has immunomodulatory effects, and deficiency upregulates proinflammatory gene expression [56]. Niacin, an amide of B_3_, reduces TNFα, IL-6, and IL-1β in stimulated alveolar macrophages and inhibits NF-κB activation [58]. Similarly, vitamin B7 (biotin) is also recognized as an immunoregulatory vitamin through its effects on proinflammatory cytokine expression [59].

#### 3.2.2. Mechanism of Action in Disease

Vitamins B_6_, B_12_, and folate play important and complementary roles in both innate and adaptive immune responses and have been granted health claims in the European Union for contributing to the normal function of the immune system [64,65]. It is not surprising that deficiencies in these vitamins can impair immune functions. Vitamin B_6_ reduces the function and proliferation of T-lymphocytes and inhibits cytokine/chemokine release [56]. Vitamin B_9_ (folate) deficiency has been reported to lead to megaloblastic anemia, failure to thrive, and infections due to combined immunodeficiency with an impaired T-cell proliferation response, pan-hypogammaglobinemia, and an altered proinflammatory cytokine profile, which are reversed with folate therapy [66]. Vitamin B_12_ (cobalamin) deficiency is particularly common in the elderly due to reduced absorption [67], and induces an imbalance in the cytokine and growth factor network in the CNS. Studies allude to a role in mediating the immune response to viral infection, as supplementation significantly improved sustained viral response rates in patients chronically infected with hepatitis C virus [68]. 

#### 3.2.3. Respiratory Disease

Thiamine acts as a cofactor for pyruvate dehydrogenase, the enzyme necessary for converting pyruvate to acetyl-coenzyme A (acetyl-CoA) for entry into the Krebs cycle. When thiamine levels are insufficient, pyruvate is unable to be converted to acetyl-CoA, resulting in impaired aerobic respiration and a compulsory shift to the anaerobic pathway, resulting in elevated serum lactate [69]. Thiamine and niacin is also needed for the generation of NADPH and glutathione cycling, which is an important antioxidant pathway [70]. These pathways (Figure 5) form the basis of thiamine administration in critically ill populations, which demonstrate reduced lactate and improved mortality in a trial of patients with septic shock [71]. It is of particular relevance that there are a number of studies showing a benefit of thiamine (200 mg every 12 h) in combination with vitamin C (1500 mg every 6 h) and hydrocortisone (50 mg every 6 h) in people with sepsis [60], leading to dramatic improvements in organ injury, time to shock reversal, and mortality [60] as well as severe pneumonia [72]. Fimognari et al. reported lower levels of folate and vitamin B12 in chronic obstructive pulmonary disease (COPD) patients [73], but there is little evidence of the role of supplementation on improving symptoms, hospitalization, or pulmonary function [74].

#### 3.2.4. Relevance to COVID-19

The coronavirus polyprotein encodes two proteases, called 3-C-like protease (M-pro) and a papain-like protease (PL-pro), which were previous targets for drug discovery in the SARS and MERS coronavirus epidemics [75]. A recent study utilized the available crystal structure of SARS-CoV-2 protein M-pro to screen existing approved drugs to see if they could be repurposed to combat COVID-19 [62,76]. This study, based on docking scores, ligand efficiency, lipophilic, and hydrogen bonding interactions to predict the more powerful binding drugs, found that vitamin B_12_ and nicotinamide ranked at the fourth and sixth positions, respectively [62]. Another computational study also investigated molecules that are predicted to bind tightly to M-pro in SARS-CoV-2 and identified that folate has the potential to form strong hydrogen bonds with active site residues and therefore be a possible therapeutic strategy [63]. These computational screening tools may allow targeted drug testing to be undertaken using cell-based assays and clinical trials, with niacin (B_3_), folate (B_9_), and B_12_ being possible contenders. These tools are particularly important in the COVID-19 pandemic, where there is currently no targeted therapeutics and effective treatment options remain very limited.

### 3.3. Vitamin C

#### 3.3.1. Source and Physiological Role

Vitamin C, also known as ascorbic acid (Figure 6), is an essential water-soluble nutrient, required as a cofactor for a number of enzymatic reactions required in norepinephrine biosynthesis, amidation of peptide hormones, collagen hydroxylation, hypoxia-inducible factor (HIF) hydroxylation, regulation of HIF, carnitine biosynthesis, tyrosine metabolism, and histone demethylation [77]. Its effects on the immune system during infection is wide ranging and includes the promotion of phagocytosis and chemotaxis of leucocytes and development and maturation of T-lymphocytes. It also has an important homeostatic antioxidant role, whereby phagocytes import oxidized vitamin C, (dehydroascorbic acid) and reduced vitamin C (L-ascorbic acid) is regenerated in exchange [78,79]. There is also speculation that vitamin C may play a role in mediating the adrenocortical stress response, particularly in sepsis [80].

#### 3.3.2. Mechanism of Action in Disease

Vitamin C has been proposed as a treatment for respiratory infections ever since it was first synthesized in 1933 [81]. The Nobel laureate, Linus Pauling, concluded from early randomized controlled trials (RCTs) that vitamin C prevented and alleviated the common cold and as such popularized its use in the 1970s [82,83]. It is postulated to exert an antiviral effect through direct virucidal activity and augmenting interferon production whilst also having effector mechanisms in both arms of the innate and adaptive immune system [84,85,86]. The host response to viruses and bacteria include the release of reactive oxygen species (ROS) from activated phagocytes. Paradoxically, this has been shown to be harmful to host cells and in some cases is implicated in the pathogenesis of viral and bacterial infections [87,88,89]. Animal studies support a beneficial role of vitamin C in reducing the incidence and severity of bacterial and viral infections [90]. These positive effects include increased resistance of chick embryo tracheal organ cultures to infection and protecting broiler chicks against avian coronavirus [91,92]. The relationship between oxidative stress and the induction of genes integral to the systemic inflammatory response, including TNFα, IL-1, IL-8, and ICAM-1, has been shown to be mediated through activation of the nuclear transcription factor NF-κB [93,94,95]. Vitamin C has been shown to reduce inflammation and ROS via attenuation of NF-κB activation [96].

#### 3.3.3. Respiratory Disease

Vitamin C is generally considered safe; however, reported adverse effects of high-dose intravenous vitamin C (HDIVC) include oxalate kidney stone production [97,98]. Its potential benefits, low cost, and safety profile make it an attractive therapeutic candidate in treating respiratory infections. A Cochrane review examining placebo-controlled trials testing 200 mg/day or more of oral vitamin C for preventing and treating the common cold found that supplementation did not reduce incidence in the general population [99], but doses of 6–8 g/day were associated with a shorter duration and severity of common cold symptoms with 46% symptom-free within 24 hours on an 8g dose [100]. A subgroup analysis of heavy acute physical stress trials did find that it halved the incidence in this population [101,102,103,104]. Based on these findings, vitamin C may be useful for people exposed to brief periods of severe physical exercise or in high doses for those with active cold symptoms. This work has led to the investigation of vitamin C as a candidate treatment to offset the effects of inflammation and oxidative stress seen in sepsis and ARDS. Excessive generation of potent proinflammatory cytokines and chemokines soon after the onset of sepsis sets the stage for the development of multi-organ failure (MOF) [105]. The cytokine storm can result in neutrophil migration and accumulation within the lung interstitium and bronchioalveolar space and is regarded a key determinant of progression in ARDS [106]. Neutrophil extracellular traps or NETosis is a cell death pathway different from apoptosis and necrosis that traps and kills pathogens [107]. An excessive NETosis response is a maladaptive response that leads to tissue injury, organ damage, and contributes to MOF. There is evidence to suggest that vitamin C is a novel regulator of this process [108]. Furthermore, vitamin C had been shown to significantly decrease serum TNFα and IL-1β levels and increased superoxide dismutase, catalase, and glutathione levels in a rat ARDS model supporting its antioxidant effect [109]. Additionally, vitamin C also enhances lung epithelial barrier function by promoting epigenetic and transcriptional expression of protein channels at the alveolar capillary membrane that regulate alveolar fluid clearance, which include aquaporin-5, cystic fibrosis transmembrane conductance regulator, epithelial sodium channel, and the Na^+^/K^+^-ATPase pump [110].

Vitamin C bioavailability is 100% for a single oral 200 mg dose but decreases to 33% with a single 1250 mg dose in a healthy individual [111]. Additionally, inflammatory cytokines negatively regulate an isoform of the sodium-dependent vitamin C transporter (SVCT2), which results in depletion of intracellular vitamin C [112]. Limitations in bioavailability coupled with consumption of vitamin C seen in severe infection have led to the hypothesis that therapeutic plasma levels to mitigate the degree of oxidative stress generated in critically unwell patients cannot be achieved with oral administration [113,114]. In the largest trial investigating the role of high-dose (50 mg/kg every 6 h for 4 days) intravenous vitamin C in sepsis-associated ARDS, the CITRIS-ALI trial, patients in the high-dose group did not significantly improve organ dysfunction scores or markers of inflammation and vascular injury [115]. However, mortality, while a secondary outcome, was dramatically reduced among the patients receiving vitamin C. Other authors have commentated how statistics may have been used to obfuscate the true findings of the study and thus confounding factors in that trial methodology potentially exist [116]. A meta-analysis of intravenous vitamin C supplementation in the critically ill (burns, sepsis, and septic shock) patients found that it may lead to vasopressor sparing effects, a reduced duration of ITU stay, and a reduced need for mechanical ventilation in the critically ill, without affecting overall mortality [117,118]. Combination therapies exist and consist of vitamin C along with enteral omega-3 fatty acid, γ-linolenic acid, steroids, thiamine, and antioxidants [119,120]. System biology analysis has also identified vitamin C in combination with curcumin, and glycyrrhizic acid acts on multiple hub targets closely connected and associated with immune and inflammatory responses, warranting in vitro and in vivo investigation [121]. However, these other compounds potentially confound the effect of vitamin C alone and therefore are not appraised here. Other RCTs of HDIVC in progress include the Vitamin C, Thiamine, and Steroids in Sepsis (VICTAS) trial [122]. Clearly, further carefully designed RCTs are needed to evaluate the potential role of vitamin C for other outcomes in sepsis and ARDS before supplementation is considered as a standard of care.

#### 3.3.4. Relevance to COVID-19

Given the potential role of vitamin C in sepsis and ARDS, there is gathering interest of whether supplementation could be beneficial in COVD-19 [123,124,125]. Research is gathering pace, with an HDIVC cohort study in progress in Palermo, Italy [126]. Recruitment has also begun on a new clinical trial investigating vitamin C infusion for severe 2019-nCoV-infected pneumonia in Wuhan, China. This is the first RCT to test whether there is a benefit of HDIVC in COVID-19. One-hundred and forty patients are planned to be treated with a placebo or HDIVC at a dose of 24 g/day for 7 days [127]. Despite what we know about the antioxidant properties, antiviral effect, and pleiotropic function of vitamin C, whether or not there are beneficial pathophysiological mechanisms involved in the response to COVID-19 remain to be elucidated [128]. Given the favorable safety profile of vitamin C, current treatment is potentially justified with compassionate use until COVID-19 data becomes available.

### 3.4. Vitamin D

#### 3.4.1. Source and Physiological Role

Vitamin D (Figure 7) encompasses a number of fat-soluble secosteroids with a physiological role in mineral homeostasis, primarily calcium, magnesium, and phosphate. As such, deficiencies in vitamin D have been implicated in a number of metabolic bone diseases, such as osteoporosis, osteomalacia, and rickets [129]. Vitamin D in its natural form, cholecalciferol, is acquired through dietary sources, such as oily fish and egg yolks, but is also produced through de novo synthesis in the stratum basale and stratum spinosum of the epidermis using dehydrocholesterol in the presence of ultraviolet B (UVB) radiation. Cholecalciferol is thereafter hydroxylated into its biologically active forms 25-hydroxyvitamin D (calcifediol) and 1,25-dihydroxyvitamin D (calcitriol) [130].

In addition to its well-recognized roles in calcium and bone homeostasis, the physiological roles of vitamin D also include immunomodulation and in critically ill patients [131], and there are well-documented associations between vitamin D deficiency and infection rates [132], renal and respiratory failure [133,134], sepsis [135], and mortality [136]. There have been a number of studies in which an association between vitamin D deficiencies have also been associated with immunological disorders, such as multiple sclerosis [137], ulcerative colitis [138], Crohn’s disease [139], and asthma [140].

Vitamin D deficiency is believed to affect one billion people worldwide [141]. At particular risk are individuals with darker skin, as the increased levels of pigmentation hinder the penetration of ultraviolet (UV) light needed for epidermal vitamin D synthesis. Additional at-risk groups include the elderly, who are often deficient in vitamin D, compounded by increased time indoors, reduced epidermal synthesis, and impaired vitamin D metabolism with some pharmaceutical agents, such as antihypertensives and antiepileptics [142]. Furthermore, as our knowledge of the causality of UV light exposure and the development of skin cancer and skin ageing has deepened, our behaviors have necessarily adjusted to minimize our exposure to UV light, with protective clothing and solar protective factors becoming more commonplace over the course of the last century. A sun cream with a sun protection factor (SPF) of only 8 is sufficient to reduce epidermal vitamin D synthesis by 95% [143]. As vitamin D is so difficult to acquire through dietary means, and especially challenging to acquire through sunlight in the autumn and winter months, many commonplace foods are now fortified with vitamin D. Nonetheless, Public Health England recommended in 2016 that a daily supplement of 10 µg vitamin D should be considered for all during the autumn and winter months to minimize the risk, and consequences, of deficiency [144].

#### 3.4.2. Mechanism of Action in Disease

Beyond its role in mineral homeostasis, recent research has emerged regarding the potential role of vitamin D as an immune system regulator. With regard to infectious and immune etiology, vitamin D has been linked to increased severity of numerous viral infections, including human immunodeficiency virus (HIV) [145,146], and for certain bacterial infections, including TB [147]. Supplementation with vitamin D has also been reported previously as reducing the risk of acute upper respiratory tract infections, which has led to speculation that there may be a role for vitamin D in the response to COVID-19 [36,148].

It has been hypothesized that vitamin D exerts its antimicrobial effects in three main ways: Through augmenting natural protective barriers, enhancing innate cellular immunity, and boosting adaptive immunity [149]. With regard to reinforcing natural barriers, vitamin D has been implicated in the preservation of tight junctions, gap junctions, and adheren junctions between epithelial cells, the disruption of which is a pathogenic mechanism of upper respiratory tract viruses, such as respiratory syncytial virus [150,151]. Specifically, with regard to COVID-19, correction of vitamin D deficiency is thought to suppress CD26/DDP4, one of the adhesion molecules through which the closely related COVID-MERS virus and indeed the COVID-19 virus is believed to acquire access to host cells [152,153,154]. Should the virus successfully invade the host, the next immunological barrier is the innate immune system. Vitamin D has been shown to enhance innate immunity through promoting the release of defensins and cathelicidins, the latter of which have demonstrated direct antimicrobial effects against enveloped and non-enveloped viruses [155,156]. Vitamin D also has been shown, in both animal data and clinical data, to be linked to reduced viral replication [157].

Of further significance is the role of vitamin D in mediating the inflammation that underlies ALI and ARDS. Vitamin D deficiency in particular has been recognized as a direct contributor to ARDS in the aftermath of bacterial sepsis, major surgery, and non-cardiogenic respiratory failure [158,159,160]. Vitamin D is believed to stimulate Th2 cells and regulatory T cells yet attenuate Th1 cells. This leads to a reduction in the release of proinflammatory cytokines, such as TNFα and IFNγ, by Th1 cells, which is believed to contribute to the pathogenesis of ARDS [161,162,163]. Vitamin D also upregulates the expression of some antioxidant genes, such as glutathione reductase, reducing the free radicals generated in inflammation, which are believed to contribute to the pulmonary damage that leads to the development of ARDS [164].

#### 3.4.3. Respiratory Disease

Specifically, with regard to pulmonary physiology, it has been demonstrated in animal models that vitamin D attenuates microbial ALI and ARDS through modulating the expression of the renin-angiotensin system (RAS), including ACE 1 and 2 [165]. One of the key pathogenic mediators of microbial-induced ALI is the increase in alveolar capillary membrane permeability, which evokes pulmonary oedema, hypoxemia, and pulmonary hypertension. The respiratory epithelium is able to convert vitamin D to its active form as part of local paracrine and autocrine signaling pathways implicated in host defense [166]. ACE2 enzyme inactivates angiotensin II, and as such acts as a negative regulator of the RAS. As such, ACE2 has been deemed protective against the development of ARDS, with animal models suggesting a key role in regulating vascular permeability, lung oedema, and oxygenation [167]. In a rat model of ARDS, calcitriol was demonstrated to upregulate pulmonary ACE2 and downregulate renin and angiotensin II, indicating there may be a key mechanistic role for vitamin D in hindering the progression of infection-induced ARDS [165]. Results from a meta-analysis of vitamin D supplementation and risk of acute (bacterial and viral) respiratory tract infection show a 12% overall protective effect of vitamin D supplementation. This increased to 19% with a daily or weekly regimen compared to a monthly bolus regimen. Furthermore, a 70% protective effect was observed when deficiency was corrected [168].

#### 3.4.4. Relevance to COVID-19

This picture is confounded somewhat in the case of COVID-19, which acquires entry to cells through binding to ACE2 [1]. However, the binding of the viral S1 spike protein to ACE2 causes both the virus and the enzyme to be translocated into the cell through endocytosis, thereby effectively reducing the surface expression of ACE2 and possibly contributing to the progression of pulmonary disease [169]. There does appear to be associations between high levels of ACE2 and survival benefit, implicating the attenuation of the RAS system as a means of protection against ARDS [170]. Ethnic variations in the expression of ACE2 receptors have also been noted, with the highest expression seen in East Asian males [171]. The ethnic disparities in ACE2 expression and polymorphisms may be a contributor to disease severity either independently or in conjunction with vitamin D status, and warrants further investigation. Additionally, the higher preponderance of male:female sex-specific COVID-19 mortality may in part be related to hormonal dependency of expression and/or activity of ACE2 seen in animal studies [172]. The effects of severe vitamin D deficiency have been explored in humans: Following the inhalation of bacterial cell wall constituent, lipopolysaccharide (LPS), a marked increase in alveolar inflammation (IL-1B) was noted in vitamin D-deficient individuals compared to those with mild deficiency [173]. Specifically, there has been increasing speculation that vitamin D deficiency may underpin the likelihood of mortality and disease severity in COVID-19 [125,174,175,176,177,178,179,180,181,182,183,184,185,186,187,188,189]. Observed differences in COVID-19 mortality between the northern and southern hemispheres also add to the case for vitamin D having a role in the pathogenesis of COVID-19 [190]. The emergence of Brazil as a disease hotspot may change these findings in time. Crude pan-European analysis, however, showed a negative correlation between mean levels of vitamin D in each country and the number of COVID-19 cases and mortality [191,192]. These associations, along with the physiological and immunological roles of vitamin D summarized in Figure 8, have prompted clinical trials in vitamin D supplementation in COVID-19 patients and warrant further mechanistic investigation [193,194].

### 3.5. Vitamin E

#### 3.5.1. Source and Physiological Role

Vitamin E is fat-soluble compound, which consists of eight isoforms, four tocopherols (α-, β-, γ, and δ-tocopherols), and four tocotrienols (α-, β-, γ-, and δ-tocotrienols), and it is a lipid component of biological membranes. The various isoforms are not interchangeable and only α-tocopherol meets the human vitamin E requirements. The main source in human diet varies depending on the isoform, with α-tocopherol found predominantly in sources, such as nuts like almonds and hazelnuts; legumes, such as peanuts; as well as avocados and sunflower seeds.

Vitamin E is absorbed via the small intestine and metabolized, regulated, and excreted by the liver. Vitamin E as α-tocopherol, and to a lesser degree tocotrienols, is processed by the liver by α-tocopherol transfer protein into chylomicrons [195] that are transported to tissues in the lymphatic system [196]. Although α-tocopherol is the main biologically active isoform, some evidence has shown that tocotrienols have a more potent antioxidant effect in neutralizing peroxyl radicals and lipid peroxidation in rat models; however, owing to its lesser bioavailability, less than 3% of research has focused on the tocotrienol isoforms [197,198,199].

#### 3.5.2. Mechanism of Action in Disease

Vitamin E is considered a potent antioxidant capable of neutralizing free radicals and ROS by donating a hydrogen ion from its chromanol ring (Figure 9). Free radicals generated from metabolic processes react with polyunsaturated fats within the cell membrane, causing peroxidative decomposition [200]. Vitamin E deficiency results in greater levels of lipid peroxidation in both in vivo and in vitro models and this is supported clinically by an inverse relationship between plasma lipoperoxidase and vitamin E in ARDS patients [201,202,203].

Vitamin E has been shown to enhance the immune response both in animal and human models through the following mechanisms: (1) Decreased production of nitrogen oxide resulting in prostaglandin E2 downregulation and inhibition of cyclooxygenase-2, (2) initiation of T-lymphocyte signals, and (3) modulation of the Th1/Th2 balance (Figure 10) [204].

An additional immunomodulatory effect of vitamin E is actioned through protein kinase C (PKC). Inhibiting PKC has been shown to affect the proliferation of monocytes, macrophages, neutrophils, and smooth muscle cells, and reduce superoxide free radical production in neutrophils and macrophages [205].

#### 3.5.3. Respiratory Disease

Clinical trials have shown some benefit in relation to vitamin E supplementation and upper respiratory tract infections. Randomization of 617 nursing home residents receiving 200 IU per day for a year of vitamin E showed a reduced incidence of upper but not lower respiratory tract infections [206]. ARDS is characterized by the acute onset protein rich pulmonary edema. The initial phase of the pathological mechanism is referred to as the exudative phase [207]. Complement activation results in upregulation of granulocytes, releasing ROS [208]. When liberated, these ROS result in lipid peroxidation in cell membranes, resulting in increased permeability and loss of integrity. This has been demonstrated in lung parenchyma, resulting in increased protein permeability. High thiobarbituric acid reactive substances (TBARS), an indicator of plasma lipid peroxidation, and low α-tocopherol levels have been recognized previously in critically ill patients [209]. One study specifically examined the tocopherol level as related to plasma total lipid (tocopherol:lipid ratio), to control for the dependency of plasma tocopherol concentrations on the level of circulating lipids, and showed a specific deficiency in ARDS patients as compared to non-ARDS patients [210]. It is stipulated that a combination of excessive ROS formation and lipoperoxidation during ARDS and septicemia result in a consumptive loss of vitamin E due to its antioxidant effect. In a rabbit albumin microembolization ARDS model, enteral α-tocopherol in vegetable oil versus enteral vegetable oil control significantly improved lung compliance and gas exchange and demonstrated dose-dependent effects on vascular permeability and pulmonary artery pressure. Other animal models have demonstrated similar results with aerosolized α-tocopherol [211,212]. However, no randomized controlled trials were identified that met the focus of this review [213].

#### 3.5.4. Relevance to COVID-19

COVID-19, as with most viral respiratory infections, has a predilection for those that are immunosuppressed, those with chronic ailments, and the elderly. Immunosenescence refers to the gradual deterioration of the immune system with age. Vitamin E has been illustrated to enhance T lymphocyte-mediated immune function in response to mitogens and IL-2 but also neutrophil and natural killer function, the decline of which is seen with increasing age [206,214,215,216].

Oxidative stress is one of the driving pathological mechanisms that underpins the biology of ARDS as a result of COVID-19. The oxidant-antioxidant balance is severely shifted, resulting in excessive lipid peroxidation and failure of biological membranes. The diffuse alveolar damage, hyaline membrane formation, and pulmonary edema are the pathological outcomes seen in the most severely affected [217,218]. Vitamin E ingestion is known to lower the production of superoxides and perhaps tilt the balance back in favor of antioxidants. Deficiency in animal models has also been shown to cause increased genetic mutations that promote the virulence of coxsackievirus, and influenza virus, and two RNA viruses, such as COVID-19 [65,219].

As global efforts turn towards the production of a vaccine, vitamin E supplementation may enhance vaccine efficacy in those most susceptible within our society as seen with increased tetanus antibody titers [214]. Highly complex mechanisms underpin the biological effects of vitamin E and as the pandemic evolves further research may unravel the potential benefits.

## 4. Conclusions

In the absence of a vaccine, the world is eagerly awaiting a panacea of treatment options for COVID-19. In this article, we critically appraised the potential immunomodulatory, antioxidant, and antimicrobial roles of vitamins A to E. Although there is currently no evidence from completed randomized controlled trials to conclusively and specifically demonstrate a role for vitamin supplementation in the fight against COVID-19, there is strong scientific evidence, based on studies of vitamin physiology, pharmacology, and their role in clinical studies of infection and ARDS to indicate a role for vitamins in the battle against this global pandemic. In particular, disease models of a lower vitamin A concentration and increasing host susceptibility to influenza and SARS-CoV have prompted investigation into the relationship between oral supplementation with vitamin A and COVID/COVID-like viruses. Furthermore, computational screening tools is a novel approach revealing promise for targeted drug testing of B vitamins, such as folate and B_12,_ and supplementation if warranted. Vitamin C, owing to its potential role in attenuating upper respiratory tract infections, its antioxidant properties, and use as a high-dose intravenous therapy in ARDS and sepsis, may prove beneficial in COVID-19. The RCTs currently underway might indeed demonstrate a role for this vitamin in the intensive care setting. The Front Line COVID-19 Critical Care (FLCCC) Working Group released the MATH+ protocol in April 2020 and included vitamin C within its multimodal therapeutic strategy. The protocol consists of intravenous methylprednisolone, high-dose intravenous ascorbic acid, full-dose low-molecular-weight heparin and optional treatment components (including thiamine, zinc, and vitamin D) [220]. This is an early intervention protocol directed at suppressing hyperinflammation seen in COVID-19. Anecdotal experience with this regime has shown that early provision (within 6 h of admission) of MATH+ has reduced the need for mechanical ventilation and improved mortality rates within North America and China. The FLCCC working group are reporting 2 deaths in 100 patients treated with the MATH+ protocol; however, they did not compare their results to a control group. These findings are striking, but larger series and tightly defined indications will be required before widespread adoption of this treatment can be advocated. The vitamin receiving the most publicity at present is vitamin D in light of the association between disease severity and populations at risk of vitamin D deficiency, the elderly and black, Asian, and minority ethnic (BAME) populations [221]. There is certainly emerging and existing evidence to postulate a mechanism through which this vitamin might play an essential role in the fight against COVID-19, including its association with the pulmonary renin-angiotensin system. The therapeutic potential of vitamin D has already captured the attention of the scientific and medical communities as evidenced through a number of emerging clinical trials and journal articles. The interest has even percolated through to government [222], with the United Kingdom now advocating the supplementation of vitamin D for individuals in minority ethnic groups, over 65s, and those confined to life indoors [223,224,225,226]. However, UK Biobank analyses of blood calcifediol concentration and COVID-19 risk contradicts existing data and government advice. Despite the calcifediol concentration being lower in BAMEs, the study failed to demonstrate an association between calcifediol and COVID-19 infection after adjusting for potential confounders [227].

It would be unjustified to claim that vitamins are the answer to the coronavirus pandemic, but it would be fair to say that there is emerging evidence that they may play a role in either preventative measures or supportive therapy in established respiratory infections and intensive care settings. The physiology, pharmacology, and basic science behind vitamins A to E does allude to potential benefits that warrant further investigation and completion of the clinical trials, even if this translates to a need for diligent deficiency correction rather than routine mass supplementation.

The current and emerging guidance to supplement at-risk populations with vitamin D is justified given the as of yet unexplained predisposition for the elderly and BAME communities to have the most severe outcomes, potentiated by the fact that an increasing number of individuals will be confined to a life indoors during the lockdown period of the COVID-19 pandemic. Caution must, however, be exercised when recommending vitamin supplementation on a larger scale: The effects of hypervitaminosis can be severe, particularly the fat-soluble vitamins A, D, and E. Of note, hypervitaminosis is almost exclusively a product of ingesting an excess of vitamin supplements, rather than a product of vitamins acquired through normal dietary and physiological means.

The value of maintaining a diet containing a balance of vitamins seems prudent and applicable to the general population during these unprecedented times. We hope in the near future that well-designed clinical trials provide the evidence needed to determine whether the clinical value of vitamins matches the promise of their antioxidative, antimicrobial, and immunomodulatory properties.

## Figures and Tables

**Figure 1 nutrients-12-02550-f001:**
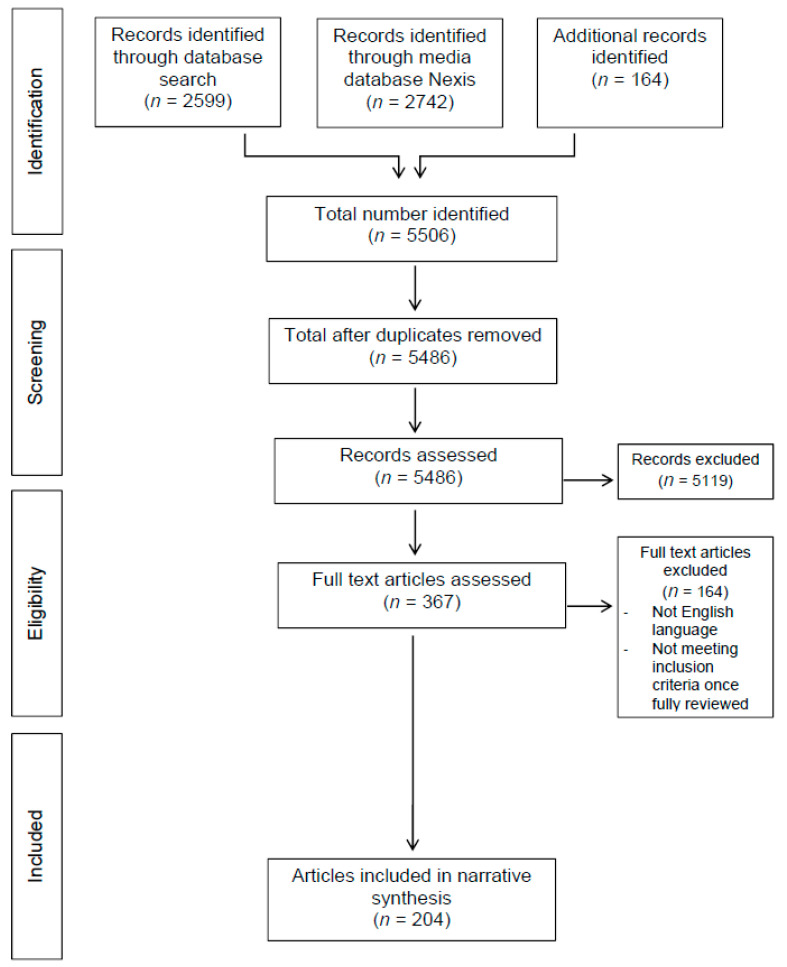
Systematic search strategy flow diagram.

**Figure 2 nutrients-12-02550-f002:**
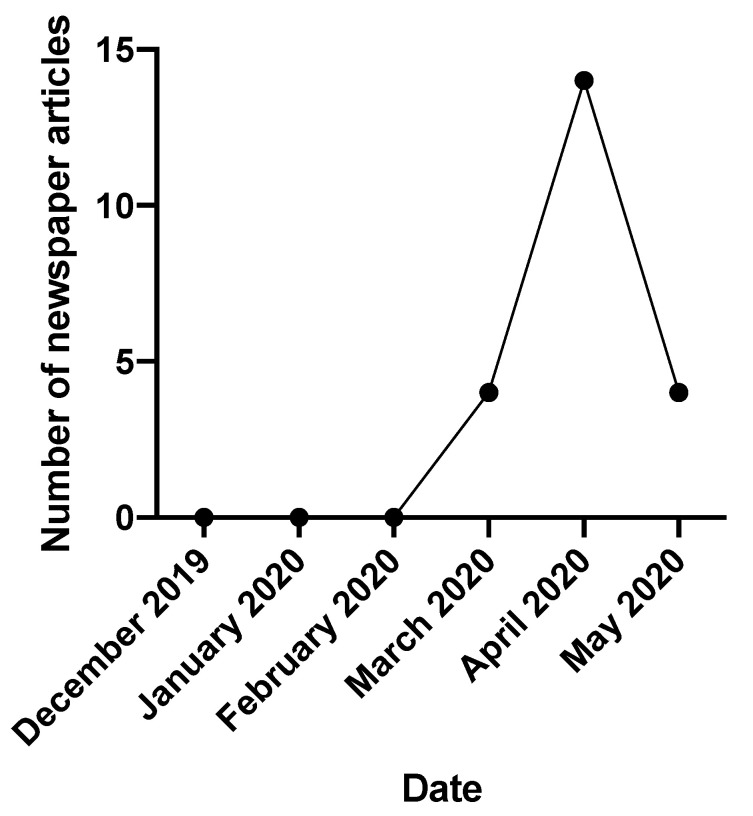
Number of newspaper articles published on vitamin supplementation in COVID-19 over time.

**Figure 3 nutrients-12-02550-f003:**
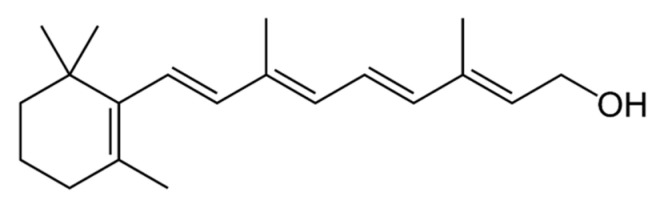
Chemical structure of vitamin A, retinol.

**Figure 4 nutrients-12-02550-f004:**
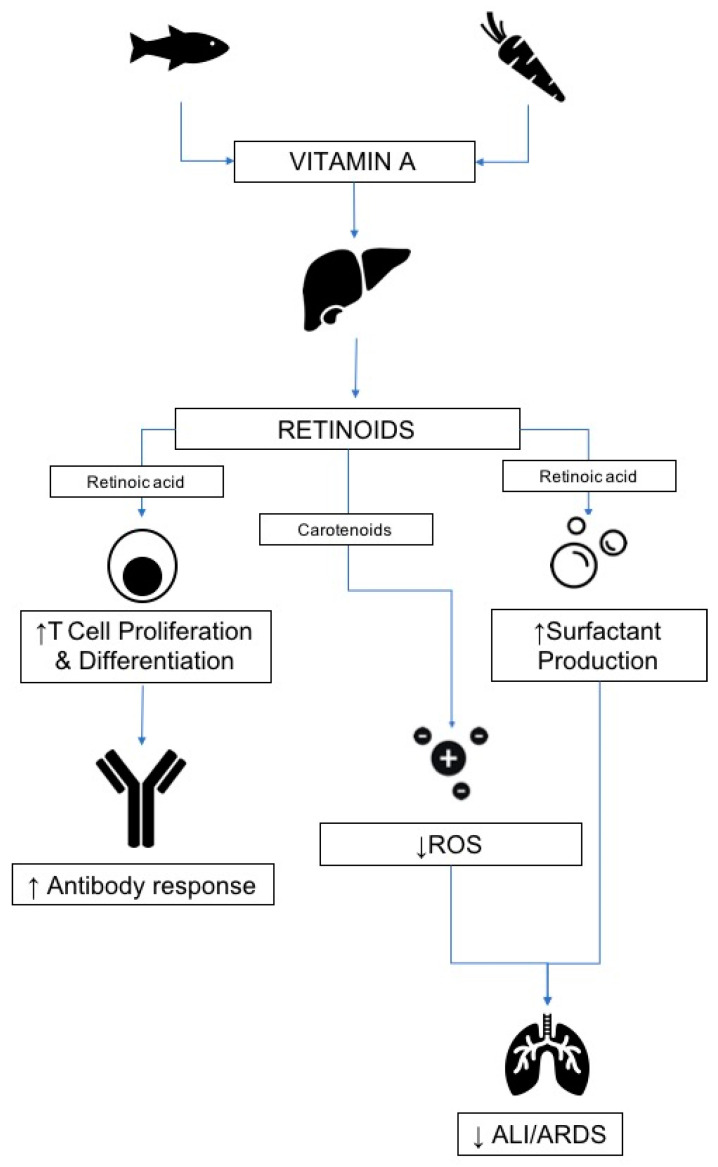
Retinoic acid and carotenoids exert a number of physiological effects, including the augmentation of T cell function, which may improve the adaptive immune response to pathogens, such as viruses. Furthermore, antioxidant and surfactant-mediating properties of vitamin A derivatives may have a protective role in the pathogenesis of ARDS, a known complication of severe cases of COVID-19.

**Figure 5 nutrients-12-02550-f005:**
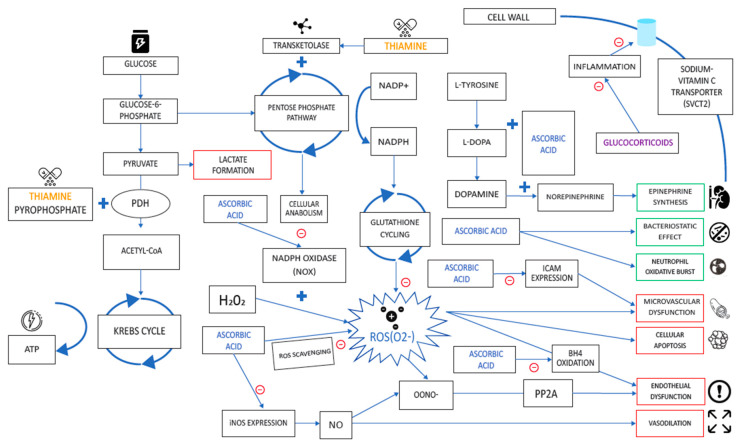
Thiamine and its derivatives act synergistically with ascorbic acid to reduce anaerobic respiration and reduce oxidative stress. It has been suggested this combination may improve mortality and organ recovery in critically ill patients with septic shock through vasoactive effects, bacteriostatic action, and immune cell mediation. The significance of these actions on viral infections, such as COVID-19, remain to be elucidated.

**Figure 6 nutrients-12-02550-f006:**
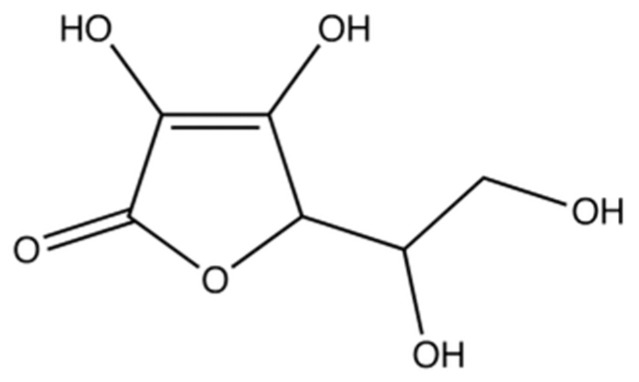
Chemical structure of vitamin C, ascorbic acid.

**Figure 7 nutrients-12-02550-f007:**
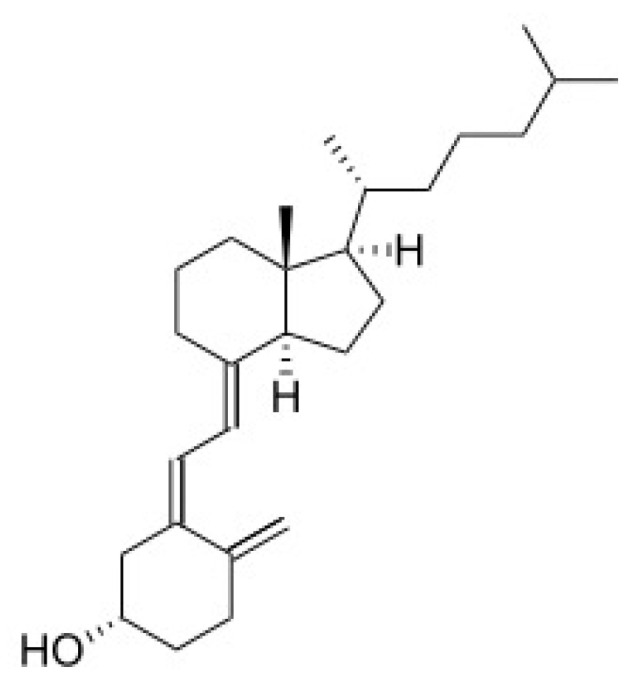
Chemical structure of vitamin D3, cholecalciferol.

**Figure 8 nutrients-12-02550-f008:**
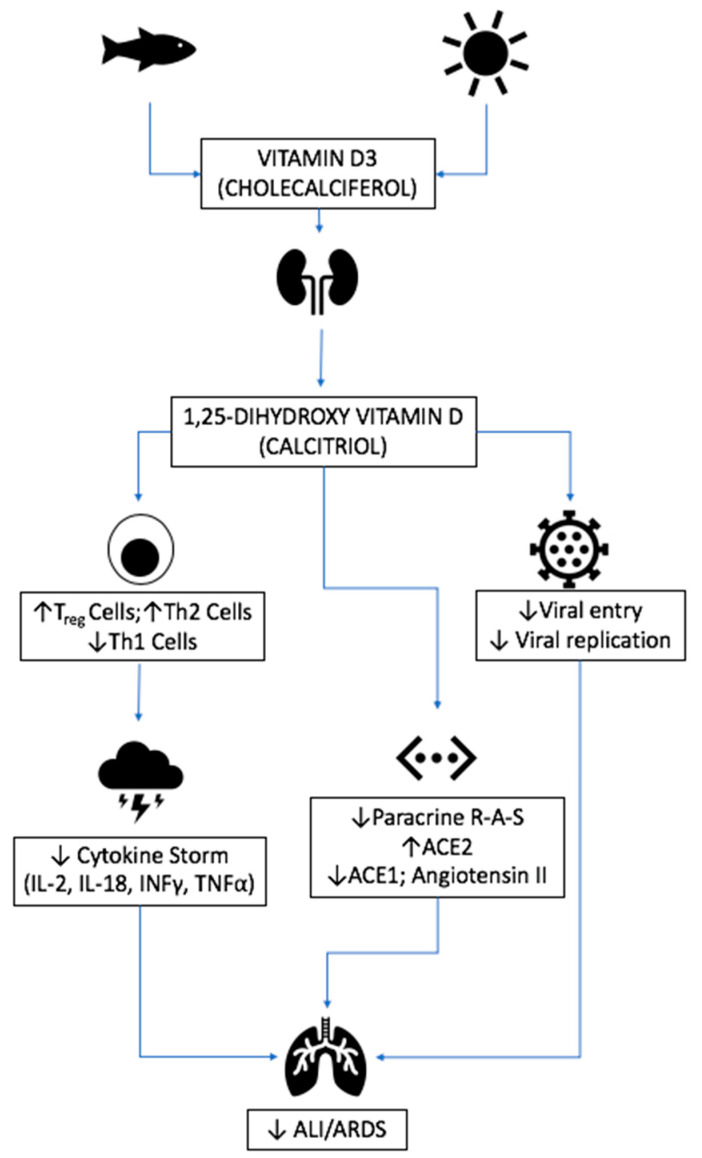
Vitamin D and potential anti-COVID19 mechanisms of action. Vitamin D derivatives are thought to mediate the immune system responses to infective agents through enhanced Treg and Th2 cell function, attenuating the cytokine storm that is thought to be a key pathogenic mechanism in ARDS. Other speculated roles include the protection against lung injury through modulating the pulmonary renin-angiotensin system and reducing viral entry and replication.

**Figure 9 nutrients-12-02550-f009:**
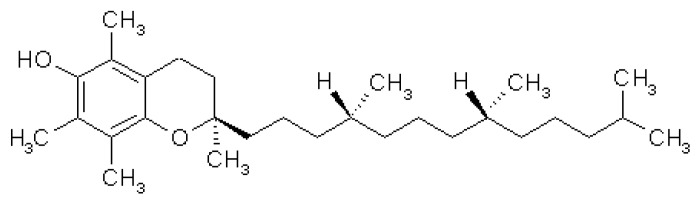
Chemical structure of vitamin E, (2′R, 4′R, 8′R)-tocopherol.

**Figure 10 nutrients-12-02550-f010:**
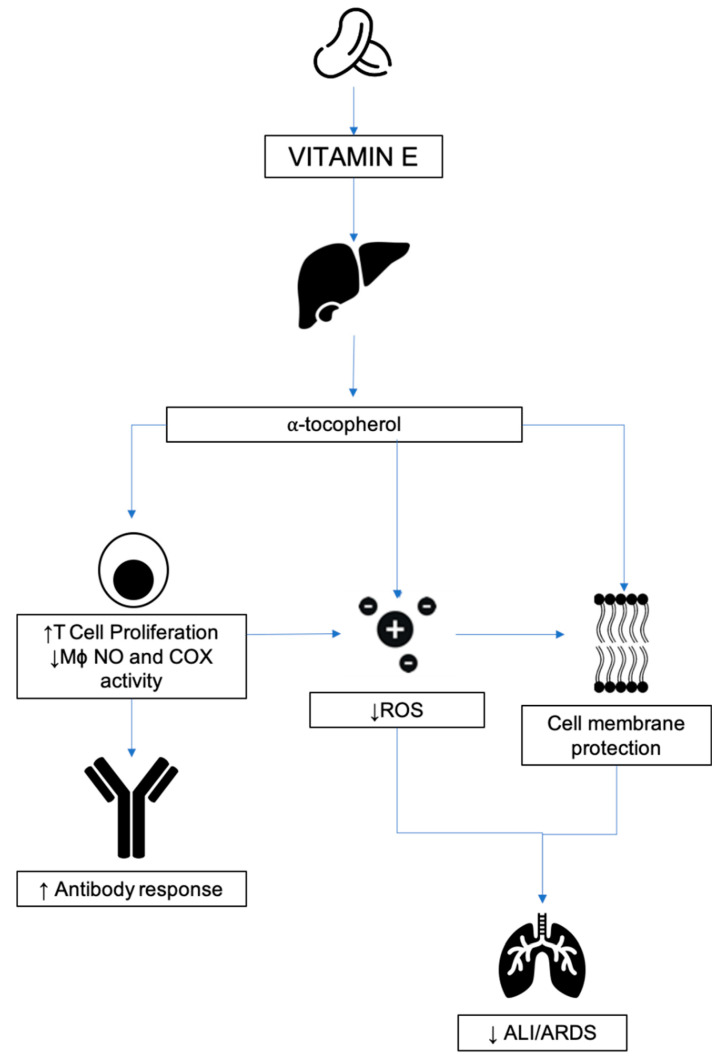
Vitamin E and potential anti-COVID19 mechanisms of action in ARDS. The antioxidant properties of Vitamin E derivatives may augment cell membrane integrity and improve the adaptive immune system response to viral respiratory tract infections.

**Table 1 nutrients-12-02550-t001:** Physiological role of the B vitamins.

B Vitamin	Chemical Name	Chemical Structure	Physiological Role	Evidence Related to SARS-CoV-2 Pandemic
B_1_	Thiamine	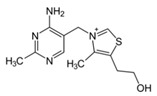	Precursor of coenzymes in sugar and amino acid catabolism	IV thiamine (together with high dose vitamin C and corticosteroids) shown to prevent deaths in people with sepsis [60]
B_2_	Riboflavin	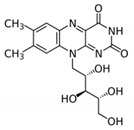	Precursor of coenzymes needed for flavoprotein enzyme reactions	Riboflavin (B2) and UV light effectively reduced the titer of MERS-CoV in human plasma [61]
B_3_	Niacin (nicotinic acid), nicotinamide, nicotinamide riboside	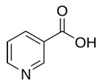	Precursor of coenzymes needed in many metabolic processes	Nicotinamide identified to have potential binding affinity for the SARS-CoV-2 protease [62]
B_5_	Pantothenic acid	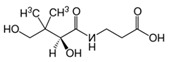	Precursor of coenzyme A	None to date
B_6_	Pyridoxine, pyridoxal, pyridoxamine	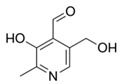	Precursor of coenzyme in metabolic reactions	None to date
B_7_	Biotin		Coenzyme for carboxylase enzymes needed for gluconeogenesis and fatty acid synthesis	None to date
B_9_	Folate	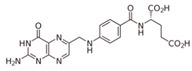	Precursor needed for DNA synthesis and repair especially during rapid cell division	Folate identified to have potential binding affinity to the SARS-CoV-2 protease [63]
B_12_	Cobalamins e.g., cyanocobalamin, methylcobalamin	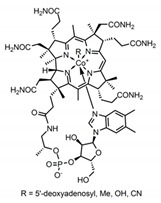	Coenzyme in metabolic reactions affecting DNA, fatty acid and amino acid metabolism	Vitamin B12 identified to have potential binding affinity to the SARS-CoV-2 protease [62]

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
