# Peer review of "Could Vitamins Help in the Fight Against COVID-19?"

_nutrients, 2020, doi:10.3390/nu12092550_

Round 1

Reviewer 1 Report

The manuscript by Jovic T. et al entitled “Could vitamins help in the fight against COVID-19?” represents a review article addressing possible interactions between vitamins respiratory disease outcomes, including recently emerging COVID-19. The authors provide systemic analysis of the observational and mechanistic literature regarding the role of different vitamins in COVID-like respiratory diseases, acute respiratory disease syndrome (ARDS) and sepsis.  Due to the current emergence and need for the COVID-19 related research, the manuscript could provide some important information as well as directions to the clinicians and investigators working in the field of respiratory diseases. However, the presented manuscript contains a number of misleading descriptions (mostly regarding vitamins, their metabolism and active metabolic forms) and statements. When describing particular vitamins and their biological role, the authors either should provide references to conventional reviews or must avoid misleading and inaccurate statements. Major concerns below should be addressed by the authors before considering this manuscript for further evaluation.

Major concerns:

  1. The exclusion criteria for 5119 records out of 5486 assessed records are not clear. The authors should provide more definitive criteria which were used during the literature inclusion into the review process.
  2. The authors’ description of vitamin A (i.e. retinols and carotenoids) is misleading and incorrect (lines 119-120) This is because the authors are trying to combine several chemically distinct compounds under term vitamin A. The authors should know that vitamin A is all-trans-retinol by definition and carotenoids are not vitamin A. The authors should introduce the term retinoids, that was coined by Sporn and colleagues in the mid-1970s (Sporn MB, Dunlop NM, Newton DL, et al. Prevention of chemical carcinogenesis by vitamin A and its synthetic analogs (retinoids). Fed Proc 1976;35:1332-8).  Retinoids comprise both natural and synthetic chemical species that bear a structural resemblance to all-trans-retinol, with or without the biological activity of vitamin A.  Hence, vitamin A (by definition all-trans-retinol) is a natural retinoid species.  Retinoids must be acquired from the diet either as preformed retinoid or as provitamin A carotenoids that are converted in the body to retinoid. Term retinoids includes other metabolites, such as retinyl esters, retinal, and retinoic acid, through which vitamin A essential functions are exerted. Retinal (11-cis-retinal) combines with opsin to form rhodopsin, the pigment in rod cells that is necessary for vision. Retinoic acid, the most active retinoid species (but not just “a common active form of vitamin A” in line 146) exerts its function through regulating the transcription of more than 500 genes upon binding to retinoic acid receptors (RARα, RARβ, and RARγ) and retinoid X receptors (RXRα, RXRβ, and RXRγ) respectively.  Moreover it is quite common to use term vitamin A when addressing nutrition, but specific retinoid species (retinol, retinoic acid) when addressing the mechanisms of action.
  3. Unlike carotenoids with their profound antioxidant properties, it is highly controversial to consider vitamin A (i.e. retinol) an antioxidant in vivo (lines 159-160), since this claim is not supported by the experimental evidence. The majority of vitamin A effects are now belived to be mediated by its effect on gene expression. Therefore the authors should clarify which effects are related to the antioxidant properties of carotenoids and which effects are exerted through regulation of gene expression by retinoic acid.
  4. Figure 4 implies that retinal itself may have an effect on T cell proliferation & differentiation as well as surfactant production, which is not supported by the literature and is inaccurate.
  5. There is inconsistency in how the authors present group B vitamins and their active (coenzyme) forms. For example, in Table 1 thiamine and pyridoxal are incorrectly named coenzymes, however thiamine diphosphate and pyridoxal phosphate are coenzyme forms of the vitamins. The same error is in line 226. In this case the authors should refer to the compounds as precursors.
  6. Line 259 – a number of enzymatic reactions involving vitamin C should be described.
  7. Line 302 – what is GULO-knockout mouse model and how its phenotype can be applied to humans?
  8. Personal correspondence with Dr. Peng cannot be considered a reliable source and included in the manuscript (line 343). Moreover building different speculations based on an unverified observation is not acceptable.
  9. Figures 4, 5, 7, 9, and 11 need figure legends providing a brief description on what the authors imply and convey.
  10. Most of the figures are too big and disproportional; figures 3, 6, 10 are too big for the information they convey and therefore should be scaled down or combined.

Typos:

Line 49 – upper respiratory tract infection (URTI)

Lines 228-229 – resulting in impaired aerobic respiration and a compulsory shift to the anaerobic pathway, resulting in elevated serum lactate.

Author Response

We thank the reviewer for their comments. We have addressed each point in the attached word document. 

Reviewer 2 Report

This was a well-written and comprehensive assessment of immunonutrition as it applies to the physiological implications for the potential use to combat the COVID-19 disease.   Although there is no concrete data to implement any immunonutritive therapies, the paper does give pause to investigate the rationales and support demonstrated in the paper.  I think it is an interesting paper, not necessarily high impact but worthy of publication in it's present form.

Author Response

We thank the reviewer for their comments.

Reviewer 3 Report

Jovic et al. summarize the potent effects of various types of vitamins to fight against COVID-19. They report the dietary source and physiological role of individual vitamin; additionally, they attempted to demonstrate the potential effects of vitamins on anti-COVID-19 by according to the known data concerning vitamin status, vitamin treatment, and respiratory symptoms. Lots of detail of the literature review is presented. In general, I found the manuscript interesting and valuable.

Author Response

(The authors gave the same response as above.)

Round 2

Reviewer 1 Report

All the comments have been addressed. Please edit line 142 for clarity.

Author Response

We have changed "Retinoids, a term coined in the mid-1970s, compromises of both natural and synthetic chemical species that all have similar structural appearances to this all-trans-retinol with or without the biological component/activity. ideals that we are familiar with for when vitamin A. is colloquially used as a term [22]" to "Retinoids, a term coined in the mid-1970s, compromises both natural and synthetic chemical species that have similar structural appearances with or without biological component/activity - the biological species being those we colloquially refer to as vitamin A [22]"